# Why, what and how do European healthcare managers use performance data? Results of a survey and workshop among members of the European Hospital and Healthcare Federation

**Damir Ivankovic** [1]*, **Mircha Poldrugovac** [1], **Pascal Garel** [2], **Niek S. Klazinga** [1], **Dionne S. Kringos** [1]

**1** Department of Public Health, Amsterdam Public Health Research Institute, Amsterdam UMC, University of Amsterdam, Amsterdam, The Netherlands, **2** The European Hospital and Healthcare Federation, Brussels, Belgium

* d.ivankovic@amsterdammumc.nl

## Abstract

Objective of this study was to better understand the use of performance data for evidence-based decision-making by managers in hospitals and other healthcare organisations in Europe in 2019. In order to explore why, what and how performance data is collected, reported and used, we conducted a cross-sectional study based on a self-reported online questionnaire and a follow-up interactive workshop. Our study population were participants of a pan-European professional Exchange Programme and their hosts (n = 125), mostly mid-level hospital managers. We found that a substantial amount of performance data is collected and reported, but could be utilised better for decision-making purposes. Motivation to collect and report performance data is equally internal and external, for improvement as well as for accountability purposes. Benchmarking between organisations is recognised as being important but is still underused. A plethora of different data sources are used, but more should be done on conceptualising, collecting, reporting and using patient-reported data. Managers working for privately owned organisations reported greater use of performance data than those working for public ones. Strategic levels of management use performance data more for justifying their decisions, while managers on operational and clinical levels use it more for day-to-day decision-making. Our study showed that, despite the substantial and increasing use of performance data for evidence-based management, there is room and need to further explore and expand its role in strategic decision-making and supporting a shift in healthcare from organisational accountability towards the model of learning organisations.

## Introduction

Despite earlier pioneering efforts [1], managing healthcare organizations by using performance data only started gaining momentum late in the 20th century on the waves of evidence-

---

**Data Availability Statement:** The data underlying the results presented in the study are available, in

the form of complete anonymised survey data sets and codebooks that do not violate privacy laws or statements applicable to this research, from the Zenodo repository; published January 14, 2020; doi: 10.5281/zenodo.3607986; https://doi.org/10.5281/zenodo.3607986.

**Funding:** This research project was funded by the Marie Skłodowska-Curie Innovative Training Network (HealthPros—Healthcare Performance Intelligence Professionals; https://www.healthpros-h2020.eu/) that has received funding from the European Union's Horizon 2020 research and innovation programme under grant agreement no. 765141. The views expressed in this manuscript are those of the authors and not necessarily those of the EU. The funders had no role in study design, data collection and analysis, decision to publish, or preparation of the manuscript. Funding to pay the Open Access publication charges for this article was provided under grant agreement no. 765141.

**Competing interests:** The authors have declared that no competing interests exist.

based medicine [2] and outcome measurement [3]. Late 1980s saw the emergence of evidence-based policy-making [4] followed by the introduction of the concept of evidence-informed decision-making and evidence-based management in the 1990s [5]. It has been argued that, in the era of accountability, work of healthcare organisations and medical practitioners is characterized by "measured performance with consequences" [6,7]. Some even go so far as to state that we already are on the verge of the post-accountability era [8].

The use of performance data has been recognised as a great opportunity to fundamentally improve the way we provide healthcare [9]. Performance data are quantitative measures used to describe the degree to which health systems and services achieve their overall and intermediate goals [10].

Measuring performance often looks at dimensions such as clinical effectiveness, efficiency, safety (of both patients and staff) and patient-centeredness, as is the case of World Health Organisation's (WHO) PATH framework for hospitals [11]. Scientific knowledge increasingly underpins the criteria for measuring performance [12] such as face validity, reproducibility, acceptability, feasibility, reliability, sensitivity and predictive validity in order to maximise the quality of indicators [13–17]. Working with performance data means monitoring (by choosing and collecting), evaluating (by analysing), and communicating (by reporting) how various components of health systems and services meet their goals and objectives to ultimately act upon performance data to improve outcomes.

It is estimated that only 50–60% of care is delivered in line with evidence and guidelines, that around a third of our medical spending has no measurable effects or justification and that little improvement has been made to the rate of adverse events (one in 10 patients) across healthcare in the past 25 years [18]. Nevertheless, some progress is visible. Healthcare systems and organisations are gradually shifting from volume-based models to more outcome-based ones [19] and they are strengthening the role of public reporting on performance [20]. This increases the pressure on healthcare organisations to monitor, evaluate and communicate performance data and is particularly true for knowledge and technology intensive healthcare organizations such as hospitals. These requirements are often mandated externally and, for most part, relate to accountability and quality assurance actions—the so-called summative approaches to performance intelligence [21]. Whether summative approaches actually lead to formative actions—quality improvement itself—is still unclear [22]. This has become increasingly relevant with the recent interest in knowledge-based health care services and systems and the strive for learning health care systems [23].

Managing hospitals, and other healthcare organisations, requires a delicate combination of strategic and operational management of clinical and all other processes that provide support for clinical work. Management, in its essence, takes place through three main processes: planning, decision-making and controlling. Performance data provide the evidence necessary to carry out all of these functions. In the field of health services management, evidence-informed management is described as "the systematic application of the best available evidence to management decision-making, aimed at improving the performance of health service organisations" [24]. Performance data are a fundamental component of the "best available evidence" and hence are essential for evidence-informed management.

Management of clinical and support processes is often the domain of middle management, linking the worlds of evidence-based clinical medicine to evidence-based management of healthcare delivery [25]. Working in data-intensive healthcare organisations like hospitals, poses a set of specific managerial challenges, especially for middle managers. They supervise frontline clinical workers and are themselves being supervised by organization's senior managers. They are thus expected to align and cascade the organization's strategic goals and promote the implementation of services delivery innovations to improve performance [26]. Research

on middle managers' commitment to the implementation of innovations shows that it is, in large part, influenced by personal perception of the potential benefit of the innovation for patients and the ease with which an innovation can be implemented [27]. Knowledge on current practices in collecting, reporting and the use of performance data by healthcare organisation management is scarce, which may hinder using the full potential of performance data. Research to date is predominantly limited to the use of performance indicators for summative purposes (hospital quality assurance and accountability), and the importance of hospital information systems for performance measurement. Research is also often limited to single-country studies [28,29].

This study aims to explore the actual use of performance data in hospitals and other healthcare organisations in Europe, and opportunities to enhance its use, by understanding (i) why is performance data collected, reported and used, (ii) what data is collected, reported and used for performance management, and (iii) how is performance data used for decision-making in healthcare organisations?

## Methods

This paper presents a descriptive cross-sectional study based on a survey, delivered through an online self-reported questionnaire, and a follow-up interactive workshop. The questionnaire was distributed to managers of hospitals and other healthcare organisations in a purposive sample of participants to the European Hospital and Healthcare Federation's (HOPE) Exchange Programme, eliciting information on the actual use of performance data in hospitals and other healthcare organisations in Europe in 2019. Survey results were presented and additionally discussed through real-time polling and a panel discussion during a workshop at the annual HOPE Agora Conference and Meetings (HOPE Agora). Most of the respondents to the survey also attended the workshop. The questionnaire and workshop results were analysed using three overarching questions: "why", "what" and "how" [30].

### Questionnaire design and piloting

The questionnaire was developed by the authors and based on their experience in the field of health system performance measurement and management. Additionally, literature on the uptake of performance data in decision-making and a survey published by the New England Journal of Medicine Catalyst project were used [31]. It consisted of open- and closed-ended questions, structured in four parts as follows: (i) professional background and managerial competencies and experience (15 items); actual use of performance data at (ii) organisational level (9 items) and (iii) in daily work of respondents (7 items); (iv) experience and opinions on enablers and barriers related to use of performance data (13 items).

The survey questions were tested [32] via two piloting streams. First, face-to-face cognitive testing interviews were performed with seven operational-level hospital managers in four Dutch hospitals. The goal of this piloting stream was to assess whether respondents were able to comprehend questions, retrieve information from memory, summarize information and report an answer [33]. Following cognitive testing, we excluded three questions, clarified and simplified the language used throughout the questionnaire and gave respondents an option of replying to open-ended questions in their native language, while the main language of the questionnaire remained English. The second testing stream included sending out an online questionnaire to HOPE National Coordinators (contact persons) in 30 HOPE member countries in Europe. The pilot version of the questionnaire also included an additional section asking about clarity and relevance of individual questions and the questionnaire in general. We received 13 completed piloting questionnaire responses from 10 countries, based on which we

amended the survey's introductory text and added a "Do not know / Not applicable" answer options to some questions. On average, piloting subjects in this stream perceived the questionnaire to be "very relevant" and "very clear".

S1 Appendix "Online questionnaire" presents the final version of the questionnaire, which totalled 44 items. It was set up as an online questionnaire using LimeSurvey online surveying platform (version 2.6.7) [34].

## Study population and data collection

The primary study population were all participants of the 2019 HOPE Exchange Programme (n = 123; from 23 countries). This is a voluntary professional exchange and training programme organised around a specific theme, for professionals with managerial responsibilities working in hospitals and other healthcare facilities. In 2019 the theme of the HOPE Exchange Programme was "evidence-informed decision-making in healthcare management". During the course of four weeks, participants move from their country of work to one of 30 HOPE member countries where they visit multiple healthcare facilities. The aim is to facilitate better understanding of health system functioning, hospital managerial work and exchange best practices across Europe. After the four-week exchange, all participants participate in an annual two-day HOPE Agora, which is also attended by the alumni of the Exchange Programme, local hosts and other stakeholders. During the HOPE Agora, the HOPE Exchange Programme participants present and discuss their study-visit experiences.

Additionally, the local hosts and alumni (2015–2018) of the Exchange Programme were also included in the study population (n = 1334) and invited to participate. All communication with the sampled study population was done by HOPE Central Office using its official mailing list. One month before receiving the questionnaire, potential study participants were sent a one-page document outlining the rationale, scope and timeline of the study. The invitation to participate in the survey was sent one week before the start of the 2019 Exchange Programme to 1457 active email addresses in 30 HOPE member countries in Europe. No return emails due to incorrect addresses or other technical issues were received. Two reminders were subsequently sent, and the data collection ended four weeks later, before the HOPE Agora.

## Data analysis

Data collected through the online questionnaire was analysed using univariate descriptive statistics. Analyses were conducted using the R statistical program version 3.6.1 [35]. Respondents were, for certain parts of the analysis, sub-grouped by their reported managerial position and experience, as well as the type of organisation they work for. Analysis was done on a full sample of respondents, including the primary, 2019 HOPE Exchange Programme participants, and the secondary study population, 2015–2018 HOPE Exchange Programme alumni and local hosts.

In the first part of the analysis, we looked at why performance data is used—making decisions or justifying decisions—and to what extent. We also analysed the motivation to use performance data, whether it is predominately internal or external, and is it perceived to comply, compare or improve. We compared differences in the motivation to use performance data between managerial roles, levels of experience or types of organisations. Finally, we analysed how respondents' confidence in reliability and validity of data influences its use.

For the second part of the analysis, we focused on finding out whether data being collected, reported and used was sufficient or even excessive. Here, we also looked at which types of data are collected, reported and used based on the WHO PATH model domains. We also analysed which data sources are used, with a special focus on patient-reported data.

In the final section of the analysis, we explored how performance data is being used and which tools are employed to report performance data. We were also interested in how performance data is integrated into respondents' daily work and how confident they are in their skills of using these tools, and performance data in general. Finally, we investigated the confidence in effective use of data for different purposes within respondents' organisations.

## Reflection workshop: Validation and contextualisation of results

In June 2019, we attended the HOPE Agora in Ljubljana, Slovenia and presented the preliminary results of the questionnaire. Having a unique opportunity to meet the majority of questionnaire respondents in person, we organised an interactive reflection workshop during on the use of performance data for management in healthcare organisations. The workshop was used to present and validate the questionnaire results as well as to clarify, supplement and contextualise the findings. In order to do so, we used real-time audience polling method [36] and organised a round-table discussion based on the questionnaire and polling results. The workshop was attended by roughly 150 individuals.

For the polling, the audience was presented a series of statements with which they could agree ("Yes") or not ("No") by voting using their mobile phones. In total, 11 questions were asked, and on average 109 replies were received per question (min = 96, max = 127). This presented a vast majority of the workshop audience. For the discussion part of the workshop, following the results presentation and audience votes, we invited five HOPE Agora attendees to join the panel discussion. The panel members represented a mixture of geographical backgrounds (Denmark, Belgium, Netherlands, Slovenia and Bulgaria) and roles in healthcare organisations as well as HOPE governance structures (clinical managers, hospital- and national-level quality managers, hospital directors and HOPE National Coordinators). The panel also featured a balanced gender representation with three women and two men participating.

## Results

### Characteristics of study population

In the primary study population, we reached a 72% response rate with 88 out of the 123 HOPE Exchange Programme participants (2019 cohort), working in 20 European countries, completing the questionnaire. Additional 37 respondents completed the questionnaire from the secondary study population (HOPE Exchange Programme participants between 2015 and 2018 as well as local hosts).

The majority of respondents were female healthcare managers working in publicly-owned hospitals with an even distribution of managerial responsibilities between strategic, clinical and support-process management. Detailed characteristics of the respondents are shown in Table 1.

### Why?

**Use of performance data as evidence and/or justification.**    Nearly half of the questionnaire respondents completely or considerably agreed that thanks to using performance data, managerial decisions taken in their daily work are more evidence-informed (46%, 58/125). At the same time, more than two-thirds of participants completely or considerably agreed that decision-making based on performance data makes it easier for them as managers to explain and justify their decision (70%, 88/125). During the workshop part of the HOPE Agora, a slight majority of the audience agreed that they indeed use performance data more to justify

**Table 1. Characteristics of questionnaire respondents.**

| Respondents | | |
|---|---|---|
| **Country of work** | **N** | **%** |
| Spain | 20 | 16 |
| United Kingdom | 13 | 10 |
| The Netherlands | 10 | 8 |
| Denmark | 9 | 7 |
| Portugal | 7 | 6 |
| Other | 66 | 53 |
| **Total** | **125** | **100** |
| **Gender** | **N** | **%** |
| Female | 91 | 73 |
| Male | 34 | 27 |
| **Total** | **125** | **100** |
| **Year of HOPE Exchange Programme participation** | **N** | **%** |
| 2019 | 88 | 70 |
| 2015–2018 | 32 | 26 |
| Exchange Programme Host | 5 | 4 |
| **Total** | **125** | **100** |
| **Organisations where respondents work** | | |
| **Type of organisation** | **N** | **%** |
| Hospital–smaller local | 21 | 17 |
| Hospital–larger regional / teaching | 39 | 31 |
| Hospital–university | 37 | 30 |
| Long-term care | 3 | 2 |
| Mental care | 4 | 3 |
| Primary care | 6 | 5 |
| Central government (ministry, agencies) | 4 | 3 |
| Local and regional government | 10 | 8 |
| Other | 1 | 1 |
| **Sub-total hospitals** | **97** | **78** |
| **Total** | **125** | **100** |
| **Organisation ownership** | **N** | **%** |
| Public | 101 | 81 |
| Private not-for-profit | 12 | 10 |
| Private for-profit | 11 | 9 |
| Other | 1 | 1 |
| **Total** | **125** | **100** |
| **Respondents' work positions** | | |
| **Management responsibilities** | **N** | **%** |
| Planning and strategy for the whole organisation | 32 | 26 |
| Care processes | 37 | 30 |
| Support of care processes | 34 | 27 |
| Not applicable | 22 | 18 |
| **Total** | **125** | **100** |
| **Number of people managed** | **N** | **%** |
| 1–5 | 14 | 14 |
| 6–20 | 27 | 26 |
| 21–50 | 14 | 14 |

(*Continued*)

**Table 1.** (Continued)

| Respondents | | |
|---|---|---|
| 51–200 | 22 | 21 |
| >200 | 8 | 8 |
| Not applicable | 18 | 17 |
| **Total** | **103** | **100** |
| **Years of managerial experience** | **N** | **%** |
| Less than 5 | 28 | 27 |
| 5–10 | 28 | 27 |
| 11–20 | 35 | 34 |
| More than 20 | 11 | 11 |
| Not applicable | 1 | 1 |
| **Total** | **103** | **100** |

Respondents self-assessed their managerial profiles to be considerably based on professional characteristics, such as experience and past results, knowledge and training and social skills and influence.

than to inform their managerial decisions (57%; 71/125). Accordingly, panellists presented their experience, pointing out that care-process managers (including the front-line clinical staff) predominantly use performance data to inform their decision-making, while the managers on the strategic level mostly use data to justify their decisions.

**Both internal and external motivation.** When asked about internal (assurance and improvement) and external (accountability and benchmarking) motivation to collect and report performance data, respondents found all of these to be important motivators (Fig 1). For all the categories, motivation was perceived somewhat higher on the level of the organisation as a whole, rather than in individual respondents' daily work, but none were statistically significant.

**Importance of benchmarking.** External comparison (benchmarking with other comparable organisations) was reported to be somewhat less important compared to other motivators to work with performance data. During the reflection workshop, less than a half of attendees replied that they learn more from monitoring their own performance over time, than from comparing it with others (44%; 55/126).

In the questionnaire, external reporting of performance was perceived as being useful for improvement, with 55% agreeing completely or considerably with this statement (69/125). Nevertheless, almost two-thirds of organisations (64%, 67/104) reported to use benchmarking only moderately, slightly or not at all.

When asked to provide examples of their organisation's use of performance data for benchmarking, some of the replies were:

"We share data and receive results comparing our data with national or regional data so we know how different or similar we are doing in it." (A department manager at a university hospital in Spain)

"In Denmark, there are 8 national quality improvement goals that all hospitals measure and compare." (Chief consultant in a university hospital in Denmark)

"Surgical safety. Rate of falls. Pressure ulcer rate." (Rehabilitation nurse specialist at a university hospital in Portugal)

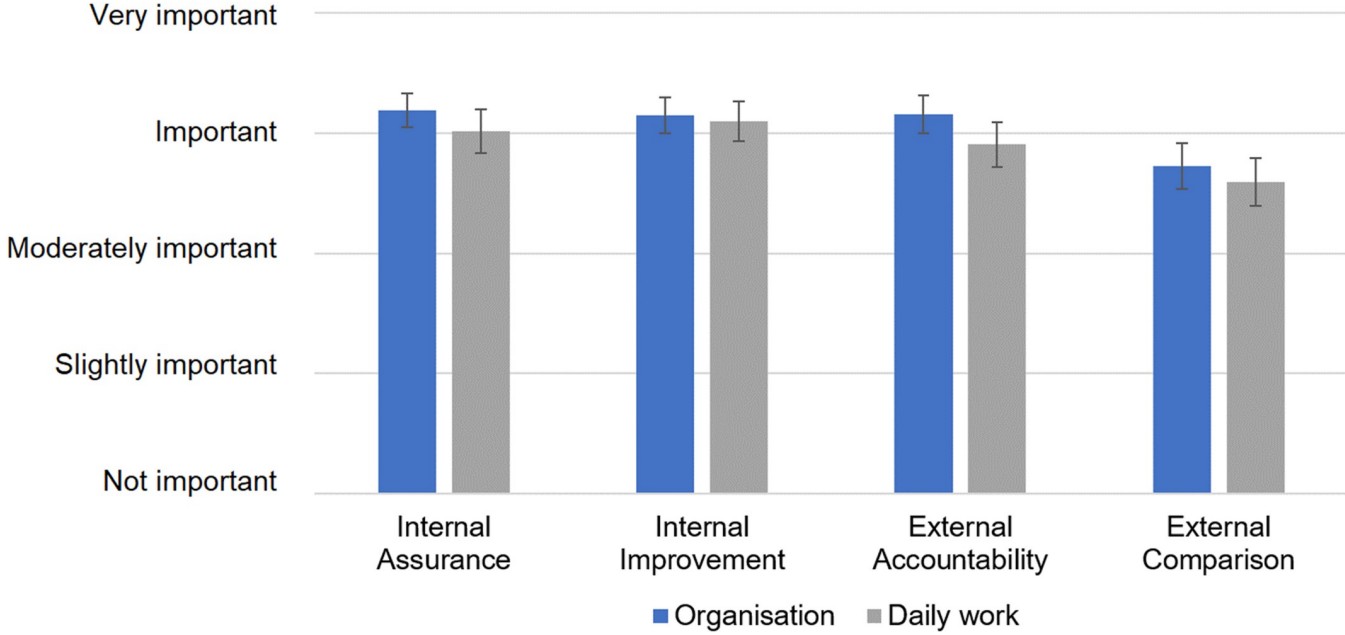

**Fig 1. Perceived importance of different motivations to collect and report performance data in respondents' organisations and their daily work.** Recoded Likert-scale reply means with 95% confidence intervals shown; for detailed results please refer to S1 Table: "Statistical data".

"Director of our hospital is in constant touch with our National Health Fund and directors of other hospitals." (Department head in a university hospital in Poland)

During the panel discussion, benchmarking was generally recognised as being useful for finding best practices and learning from others when they perform better or, as one panellist put it *"enlightening hospitals about the results of their peers so they can learn from each other"*.

**Ownership structure and motivation to collect, report and use performance data.** We found no significant differences in motivation levels between different management roles, level of managerial experience or types of healthcare organisations that the respondents work for. Across all categories, respondents working in private-for-profit healthcare organisations reported a higher level of motivation to work with performance data, but the sample size was too small (n = 11) to test significance. Hence, this finding was re-examined during the reflection workshop, where a majority of the audience indeed shared the opinion that privately-owned for-profit organisations are more motivated to use performance data (81%, 101/125). This finding was explained by private-for-profit organisations having more financial pressure to show results, while the state tends to compensate for losses accumulated by the public sector hospitals. Participants also commented that, unlike most private-for-profit organisations, the public sector often does not have adequate managerial tools at hand to incentivise better performance.

**Confidence in data and its use.** The questionnaire respondents were also asked about confidence in the validity and reliability of performance data that they use, as well as about evidence of data manipulation. A half felt completely or considerably confident in the reliability of performance data in their organisation (50%, 62/125) and 60% reported thinking that the performance indicators used in their work are valid (75/125). 16% of respondents (19/125) reported moderate or considerable evidence of data manipulation that resulted in lack of trust towards performance intelligence coming from this data.

During the workshop, a large majority replied that they would increase their use of performance data for decision-making if they had more confidence in the data (86%, 109/127) while the panel discussion showed a substantial variation in the level of confidence in data among healthcare managers present. In general, they agreed that, when looking at the healthcare system more broadly, hospitals have more reliable data, compared to, for instance, primary or long-term care providers' data.

## What?

**Balance between data collection and its use.**   Most questionnaire respondents agreed completely, considerably or moderately (82%, 103/125) that a large amount of data is currently collected, but little is used. This was supported by some of the free-text replies including:

*"24-hour discharge communication follow-up of patients, which is when they are most at risk. CQC ["Care Quality Commission"; author's comment] inspection feedback—data is public. National Staff survey. National patient survey. We have a business system, patient records system and national staff records system. We probably have an overwhelming amount of data which could be more effective if streamlined."* (Human resources professional at a mental healthcare organisation in the United Kingdom)

*"My organisation is very data rich—monitoring data on quality, safety, efficiency—for example the length of stay in a hospital, performance metrics, adherence to best practice guidance and many, many more. What is sometimes missing is the soft intelligence."* (Performance and improvement professional in the United Kingdom)

The workshop polling confirmed this finding, with 56% of the audience replying that they collect sufficient amounts of data (55/98) and 98% thinking that the existing data should be used better (100/102). The panel members discussed the actionability of data that is being collected. Despite collecting large amounts of data, some panellists argued that data itself rarely managed to answer the questions on the clinical and on the organisational level in their daily work.

Respondents were asked about the extent of their organisation's data collection, reporting and use efforts based on the domains described in the WHO PATH performance assessment system. Fig 2 shows that, across the domains, a considerable amount of performance data is collected, but somewhat less gets reported and even less is used for decision-making. Also, patient-reported data is lagging behind in terms of collection, reporting and use.

**Data collection sources and methods.**   The questions on data sources gave an insight on the variety of sources that feed into evidence-based management practices. Fig 3 shows that administrative data, and electronic health and medical records (EHRs and EMRs) data are being used considerably, while patient-reported data is being used only moderately. The workshop polling confirmed the need for collecting more patient-reported data, with 96% of the audience replying that this should be the case (98/102). Additionally, the majority agreed that ideally all patient data should be recorded in a single EHR/EMR (90%, 91/101).

The collection of patient-reported experience and outcome measures (PREMs and PROMs), and their integration into EHRs and EMRs was discussed during the workshop as a valuable tool in focusing improvement efforts and including patients' perspective of what is a good outcome. Besides including patient-reported data into medical records, the discussion in some countries now also revolves around integration of wearable-device and mobile-phone data into EHRs and EMRs.

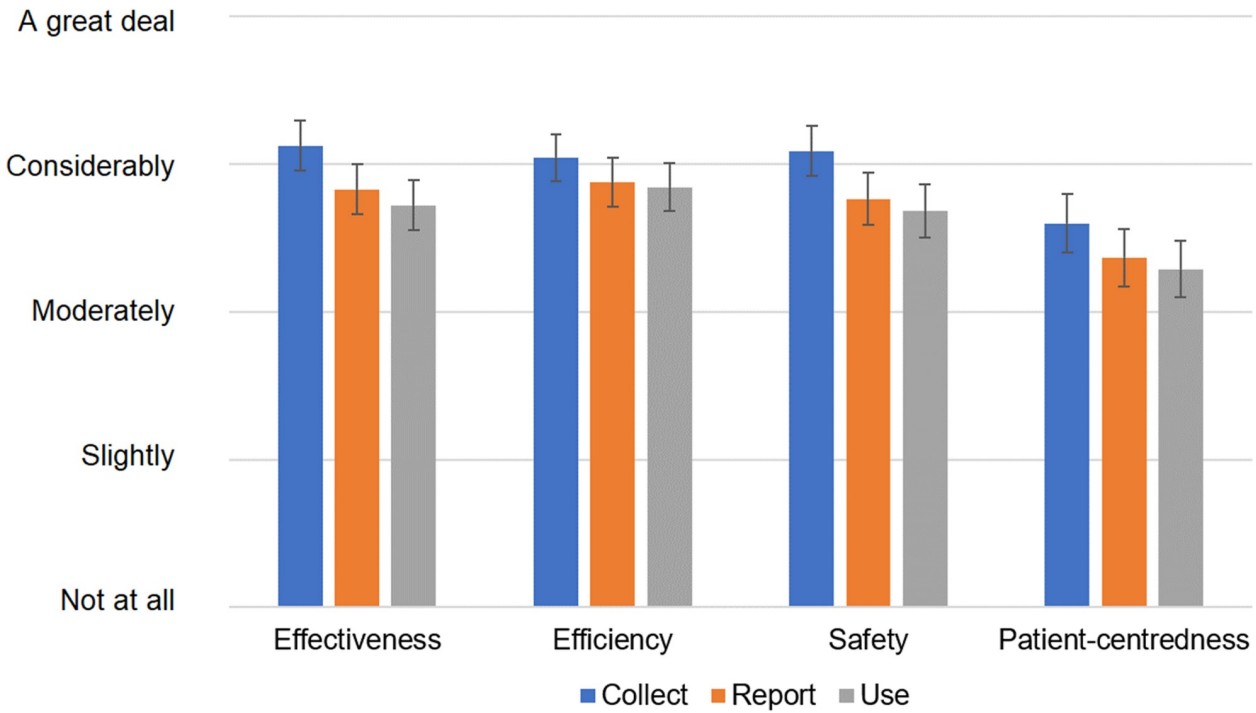

**Fig 2. Extent of data collection, reporting and usage efforts in respondents' organisations per WHO PATH framework data domains.** Recoded Likert-scale reply means with 95% confidence intervals shown; for detailed results please refer to S1 Table: "Statistical data".

Also, during the panel discussion, data collection methods have been recognised as an important improvement area, especially among Central and Eastern European countries,

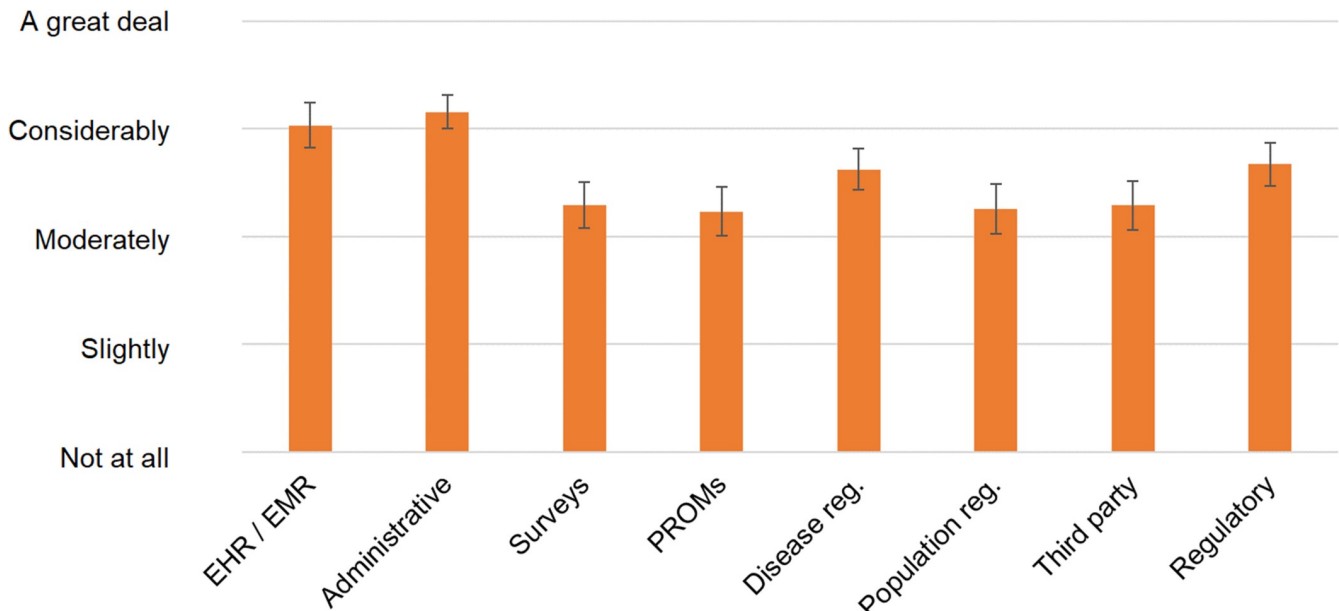

**Fig 3. Extent of use of data sources for performance data work in respondents' organisations.** Recoded Likert-scale reply means with 95% confidence intervals shown; for detailed results please refer to S1 Table: "Statistical data".

where a lot of data is still being collected manually. This was suggested to present a huge data burden and pose a risk for accuracy, quality and reliability of the data. Improving information systems was outlined as a strategy to tackle this issue.

## How?

**Reporting methods.**   Written reports were the most commonly used methods to report performance data in respondents' organisations with 86% using them (107/125). Other methods were also commonly used, such as dashboards (80/125), control charts (76/125), verbal reporting (75/125) and score cards (73/125). Almost all respondents acknowledged the use of multiple reporting methods in their organisations.

**Collecting, reporting and using performance data.**   In their daily work, the questionnaire respondents reported moderate involvement with data collection and reporting efforts and, a higher, considerable involvement with using performance data for decision-making.

This was reinforced by free text replies such as:

*"In my daily work, I collect a small part of data because the software I work with has a wide part dedicated for automated collections of data and the central office can extract them every time it's needed."* (A nurse in a regional public health authority in Italy)

*"I mainly use data that has been automatically collected and analysed. I personally undertake surveys, for example on the provision of ambulatory care services by trusts. The aim is to see whether the service arrangements are in line with the national service specifications."* (Senior regional performance lead in the United Kingdom)

**Use of performance data on operational and strategic levels of management.**   According to the questionnaire results, including the free text replies, the operational level of management (managing care and support processes) relies on using the performance data for decision-making significantly more than the strategic managerial level does:

*"In general, I would say that the use of evidence-based decision making is more prominent in process-related issues, and not so much in issues related to broader strategic issues, or organizational set-up, or team related topics. It seems easier to apply to process topics."* (Assistant to the CEO in a local hospital in Germany)

This was confirmed during the workshop with 62% of respondents replying that performance data is being used more for process (operational) management than for strategic management (61/98). Panellists discussed these results and argued that using performance data for strategic management and monitoring required building up dedicated tools and populating them with indicators linked to strategic goals. They also affirmed that the current indicator mix for process monitoring was quite good, but that indicator toolbox for strategic management needed to be built up.

**Confidence in data skills and competencies.**   A half of questionnaire respondents felt completely or considerably confident in their skills of using data (49%, 61/125). Also, around a half of them felt that funding only permits to capture and report the data, but that healthcare organisations lack the needed time and knowledge to analyse the data and use it for decision-making (48%, 60/125).

Free-text replies demonstrated that performance data mostly gets used for staffing purposes, access to care management (waiting times) and safety work (adverse events):

*"We use data to optimize processes and to monitor how we are doing. So, if we need more out-patient appointments because the waiting time gets too long, we make the decision to arrange that."* (Department manager in a teaching hospital in The Netherlands)

*"At the moment we are implementing lean healthcare in the emergency department and we are continually improving, and changing decisions, based on performance data. The same with adequating medical and nursery staff and observing results in quality assistance."* (Chief administrative officer at a local hospital in Spain)

"If there are many unexpected adverse events concerning the same topic, I will develop an action plan and I will assess the result after." (Senior manager in a French teaching hospital)

*"Depending on the clinical activity, we may need to increase capacity of clinics."* (Service manager in a teaching hospital in the United Kingdom)

*"The decision of start a new consult to reduce excessive waiting times. To analysed why in a precise moment demand of a service suddenly increased."* (Department chief in a regional hospital in Spain)

During the workshop, a vast majority of the audience confirmed that they would find it useful to improve their competences in working with performance data (93%, 89/96). Professional organisations at a national and European level were recognised as one of the stakeholders that should facilitate capacity building for the use of performance data among managers in Europe (79% replied "Yes", 80/101). During the discussion, a point was also made that healthcare managers should improve their knowledge in using data visualisation, such as process control charts.

*"Using these tools brings about a completely different discussion within the organisation. Providing historical data and showing outliers brings a much more productive discussion with clinicians. Going beyond using only charts and numbers, and visualising trend data, makes improvement more visible; both when discussing with strategic and clinical managers. Not every change is an improvement. Using these tools, they [clinicians and top management; author's comment] can see that."* (HOPE Agora panellist from Slovenia)

**Belief in organisations' effective use of data for leadership.**    The respondents were asked to assess how effective they find their organisations' use of data for business and clinical leadership, as well as for population health and individual care management. The majority found their organisations to be effective (including "very effective" and "extremely effective") in using data for guiding business (79%, 99/125) and clinical leadership (78%, 98/125). About a half of them found their organisations to be effective (including "very effective" and "extremely effective") in using data for guiding population health efforts (52%; 65/125), and two-thirds for supporting care decisions for individual patients (69%; 86/125). Compared to the NEJM Catalyst survey among the US healthcare managers, this presented a significantly higher regard for organisational effectiveness among European managers, except for population health where the results were similar. Detailed results of this question and comparison with a New England Journal of Medicine's Catalyst survey results are presented in S1 Fig: "NEJM Catalyst questions, results and comparisons".

## Discussion

With this study, we aim to improve our understanding of the status quo in performance data work among healthcare managers in Europe in 2019. We focused on mid-level managerial

staff working in hospitals and were specifically interested in learning to what extent (why, what and how) performance data gets used for decision-making. Our particular interest towards this population stemmed from the characteristics of the purposive sample of HOPE Exchange Programme participants, but also from the recognised strategic value of mid-management in supporting and implementing organisational change [37]. Middle managers are a crucial link between formative (improvement) and summative (accountability) functions of working with performance data.

We found that although a substantial amount of performance data is being regularly collected, its potential is still somewhat underused for decision-making purposes. A very similar issue is recognised in benchmarking: while being recognised as valuable, benchmarking between and within the organisations is still fairly underused. Additionally, in collecting and reporting the performance data, motivation is found both internally and externally, and is aimed at both improvement as well as the accountability purposes. Furthermore, even though there is a use of a wide range of data sources, more should be done on conceptualising, collecting, reporting and using patient-reported data. When it comes to organisations' ownership, managers working for privately-owned organisations reported a greater use of performance data compared to the ones working in the public organisations. Moreover, the strategic levels of management are reported to mostly use performance data in order to justify their decisions, while the managers working on the operational and clinical level predominantly use it for day-to-day operational decision-making.

The strength of this research work lies in it being, to our knowledge, a first Europe-wide attempt at studying the use of performance data for decision-making among healthcare managers. We reached a satisfactory response rate among our primary study population and benefited from the collaborative nature of this research work. On the other hand, a notable shortcoming of this work is the fact that the research results are limited to the HOPE Exchange Programme participants and are not generalisable to healthcare managers in Europe. Also, due to purposive sampling, sample size and distribution of responses among countries, it was not possible to conduct neither national nor regional groupings or comparisons. Due to the same reasons, we focused on data analysis using univariate descriptive statistics.

A considerable body of research has previously focused on the importance and potential usefulness of performance data in healthcare [38–41] but also on challenges and potential pitfalls of its use [42,43,44]. Our findings are no exception as they show that performance data is perceived as a potentially very helpful managerial resource, but currently provides only a moderately useful tool for managers in healthcare organisations. Only around a half of the respondents currently see performance data being helpful in their day-to-day work. A majority feels that existing data should be used better and that even some additional data should be collected. This clearly shows willingness and opportunity to make performance data work better for healthcare managers.

Different levels of management use the same performance data differently. Our findings indicate that the operational level of management uses performance data more for positive, formative, quality improvement actions, while the strategic level managers seem to mostly be using performance data for summative purposes, i.e. control through assurance and accountability. Literature indicates that this might be in line with the nature of work on different levels of management [45]. Whether this is the case in comparable populations in other geographical settings is an interesting topic for further research that we plan to pursue.

Our findings also pointed to the relative underuse of benchmarking, especially as an improvement, rather than only an accountability tool. Despite the reported underuse, respondents were very aware of possible advantages of a more wide-spread use of benchmarking, including systematically approaching assessment of practice, promoting reflection and

providing an environment for (measurable) change in clinical practice, ensuring innovative practice is acknowledged, reducing repetition of efforts and resources as well as reducing fragmentation and geographical variations in care [46]. Perceived usefulness of benchmarking, but failure to use it sufficiently or properly, could be linked to, extensively researched, possible negative consequences of public reporting [47] and measurement in general [44], but also with an under-developed culture of intra- and inter-organisational learning through benchmarking [48].

Managers in privately-owned healthcare organisations work more with performance data. They are more motivated to collect, report and use performance data, which that took place during the HOPE Agora is linked to the nature of incentive structures and the extent strategic management is held accountable in these organisations. This is also linked to the previous research recognising that differences in incentive structures between privately and publicly-owned hospitals might explain different organisational performances [49]. Performance data is not very useful to managers if it is not actionable. Actionability often stems, not exclusively from data as such, but from structures surrounding the data—in this case incentive structures linked to ownership profile of healthcare organisations. Additionally, using a larger and more focused sample, it would be interesting to explore whether differences in total healthcare system financing (state-funders vs insurance-based contracting) also play a role.

We also looked into issues of ownership and trust in data. Only a minority of respondents reported evidence and concern about data manipulation. On the other hand, an overwhelming majority of participants to the workshop stated that they were not very confident in the data they have available. There are different possible explanations for this. Lower confidence seems to be linked to less developed data collection methods and less mature "data cultures" in different geographical settings. Confidence in reliability and validity of data also depends on how and how much it is used [50]. Based on the workshop discussion, we speculate that the lack of confidence in the data might be linked to validity issues. Managers are often not convinced that the data available are really appropriate and actionable metrics of what is being measured and what they need to make decision on, in their daily managerial work [51].

Compared to their US counterparts, European healthcare professionals that use performance data to inform decision-making, felt that their organisations are more effective in using data for business and clinical leadership as well as for supporting care decisions for individual patients. This is an interesting finding that should be further examined and interpreted. Especially in the light of relatively different approaches to management, decision-making and accountability, on the system, organisation and individual level, between the European and US healthcare.

Study participants consistently acknowledged that a substantial amount of data is collected but too little is used, even claiming that data itself rarely manages to answer questions that they have. Despite this, it was never, during the course of this study, indicated that less should be collected, showing that respondents were fully aware that without input, there is no output [52]. Having performance data is sometimes not sufficient as managers need skills necessary to make effective changes in processes and cultures [53]. The analysis of the "how" of performance reporting revealed other opportunities to improve on the use of performance data. In particular, participants overwhelmingly support the need to improve their competencies in working with performance data. Also, the need for more human and financial resources to analyse the data has been articulated. Most organisations used multiple reporting tools and methods, including written reports, verbal reporting but also dashboards, control charts and scorecards. Patient-reported data, new managerial incentive structures and improved visualisation skills seem to be the next frontier in working with performance data. As indicated during the workshop that took place during the HOPE Agora, a more widespread use of visual

reporting techniques would help bridge differences and engage professionals in different positions and levels to discuss performance data and possible improvement approaches, which is another important consideration in how to make better use of existing data. Reporting on performance data should also be looked from the perspective of the communication theory, including *who* said *what*, in *which* channel, *to whom* and with what *effect* [54].

To further advance the field, building on key findings of this work but also on its limitations, we propose a number of potential future research topics and methodological approaches. Looking at a more homogenous sample of healthcare managers' profiles, and including other geographical settings—beyond Europe, while accounting for the influences of characteristics of specific healthcare systems, would lead to more generalisable results. We also suggest unpacking the reasons that hinder a more wide-spread use of performance data in decision-making on the operational level and doing so by looking at the topics of the lack of confidence in (underlying) data, timeliness, appropriateness and actionability of metrics as well as the skills needed to effectively use (including communicate) the data. Additionally, exploring the drivers of reported differences between privately and publicly-owned healthcare organisations, in their use of performance data, could optimise cross-learning opportunities. When looking into skills, crucial for middle managers in order to enhance the use of data, we propose paying special attention to benchmarking, as an organisational learning tool, which currently seems underused. Based on our experience from this study, we also suggest study designs and research modalities that do not only focus on the ability to ask questions. In line with (participatory) action research methodologies, future research should also try to engage participants into a discussion, validating survey-based data and eliciting further contextual information and even disagreements, thus deepening our understanding of what drives effective and efficient use of performance data for decision-making.

## Conclusion

There is a significant momentum among managers in European healthcare organisations to use and simultaneously improve the use of performance data for decision-making. Recognised supportive strategies are improving ownership of the data, enhancing competencies of managers in harvesting the potential use of available performance data and dedicating sufficient resources to making the best out of what is already collected. Additionally, working on improving how performance data are visually presented and communicated to managers and staff, and how benchmarking gets conceptualised and employed will be crucial strategies to increase the use performance data in the future. There is a clear need for a shift in the approach from performance data being used solely as an accountability and scrutiny measure towards one which is less focused on mandatory measurements, and is, in contrast, prioritising the use of data for decision-making, e.g. improvement science. This would also help the voices of patients, for whom our healthcare systems exist, being heard better. In this transformation, from data-driven accountable towards data-driven learning organisations, special attention should be given to optimising the role of middle management as an organisational linking pin. Their—often extensive and diverse—work experience and profound knowledge of the functioning of different organisations' components, give them a unique position to understand, but also change day-to-day organisational issues in the scope of organisations' big picture.

## Supporting information

**S1 Appendix. Online questionnaire.** The final version of the online questionnaire.
(DOCX)

**S1 Table. Statistical data.** Full statistical data linked to figures.
(XLSX)

**S1 Fig. NEJM Catalyst questions, results and comparisons.** Comparison between the results to the same question used for the NEJM Catalyst survey among the US healthcare managers and European healthcare managers in this survey.
(TIF)

## Acknowledgments

The authors thank all the participants who took part in this research project. We would also like to thank the whole HOPE team for their support in the idea development and establishing collaboration as well as for the technical support in making the online questionnaire and workshop possible. Special thanks go to Veronique Bos for her invaluable help in piloting the questionnaire. We would additionally like to thank Maja Zdolšek for the technical help in organising the workshop in Ljubljana and to AMC Clinical Research Unit for the help in setting up the online questionnaire.

## Author Contributions

**Conceptualization:** Damir Ivankovic, Pascal Garel, Niek S. Klazinga, Dionne S. Kringos.

**Data curation:** Damir Ivankovic.

**Formal analysis:** Damir Ivankovic.

**Funding acquisition:** Niek S. Klazinga, Dionne S. Kringos.

**Investigation:** Damir Ivankovic, Mircha Poldrugovac.

**Methodology:** Damir Ivankovic, Mircha Poldrugovac, Pascal Garel, Niek S. Klazinga, Dionne S. Kringos.

**Project administration:** Damir Ivankovic, Pascal Garel, Dionne S. Kringos.

**Resources:** Damir Ivankovic, Niek S. Klazinga, Dionne S. Kringos.

**Software:** Damir Ivankovic, Mircha Poldrugovac.

**Supervision:** Niek S. Klazinga, Dionne S. Kringos.

**Validation:** Damir Ivankovic, Mircha Poldrugovac, Pascal Garel, Niek S. Klazinga, Dionne S. Kringos.

**Visualization:** Damir Ivankovic.

**Writing – original draft:** Damir Ivankovic.

**Writing – review & editing:** Damir Ivankovic, Mircha Poldrugovac, Pascal Garel, Niek S. Klazinga, Dionne S. Kringos.

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
