## [Decision Letter · Decision Letter 0]

21 Feb 2020

PONE-D-20-01236

Why, what and how do European healthcare managers use performance data? Results of a survey and workshop among members of the European Hospital and Healthcare Federation

PLOS ONE

Dear Mr Ivankovic,

Thank you for submitting your manuscript to PLOS ONE. After careful consideration, we feel that it has merit but does not fully meet PLOS ONE’s publication criteria as it currently stands. Therefore, we invite you to submit a revised version of the manuscript that addresses the points raised during the review process.

We would appreciate receiving your revised manuscript by Apr 06 2020 11:59PM. To enhance the reproducibility of your results, we recommend that if applicable you deposit your laboratory protocols in protocols.io, where a protocol can be assigned its own identifier (DOI) such that it can be cited independently in the future. For instructions see: http://journals.plos.org/plosone/s/submission-guidelines#loc-laboratory-protocols

We look forward to receiving your revised manuscript.

Kind regards,

Itamar Ashkenazi

Academic Editor

PLOS ONE

Journal Requirements:

Reviewers' comments:

Reviewer's Responses to Questions

**Comments to the Author**

1. Is the manuscript technically sound, and do the data support the conclusions?

Reviewer #1: Yes

2. Has the statistical analysis been performed appropriately and rigorously? 

Reviewer #1: N/A

3. Have the authors made all data underlying the findings in their manuscript fully available?

Reviewer #1: Yes

4. Is the manuscript presented in an intelligible fashion and written in standard English?

Reviewer #1: Yes

5. Review Comments to the Author

Reviewer #1: In this paper, the authors investigated the data for evidence-based decision-making by managers in hospitals and healthcare organization in Europe in 2019. From the discussion and conclusion, this paper emphasized the data how to be collected and used. This paper seems interesting for readers to understand Europe's managers why, what and how to use performance data. The authors may add the sections of suggestion and further study in this paper.

6. PLOS authors have the option to publish the peer review history of their article (what does this mean?). If published, this will include your full peer review and any attached files.

Reviewer #1: No

---

## [Author Response · Author response to Decision Letter 0]

11 Mar 2020

Dear Editor and Reviewer,

Thank you for giving us the opportunity to submit a revised version of the manuscript entitled: “Why, what and how do European healthcare managers use performance data? Results of a survey and workshop among members of the European Hospital and Healthcare Federation” (PONE-S-20-01560).

We greatly appreciate your comments and those of the reviewers. Please find below a point-by-point reply to the comments made.

All revisions to the manuscript have been added using track changes. A “clean” revised version of the manuscript has also been submitted, in line with journal’s requirements.

Comments from the Editor:

E1. Please ensure that your manuscript meets PLOS ONE's style requirements, including those for file naming.

We proceeded accordingly and aligned the supporting file names with PLOS ONE’s style requirements. Throughout the manuscript, when referring to the supporting files, appropriate amendments have also been made.

E2. We note that you have indicated that data from this study are available upon request. PLOS only allows data to be available upon request if there are legal or ethical restrictions on sharing data publicly. If there are ethical or legal restrictions on sharing a de-identified data set, please explain them. If there are no restrictions, please upload the minimal anonymized data set necessary to replicate your study findings.

We apologise if this was not made clear enough in the original submission. 

The data set, underlying this paper’s findings, was made freely and publicly available using a data repository service. Before doing this, and due to collection of certain personal information in the questionnaire, we sought the advice of Amsterdam UMC’s Data Privacy Officer. Following their suggestion, and prior to making it publicly available, the data set was anonymised. Nevertheless, the available published anonymised data set allows for a complete reproduction of the analysis done in the paper. The data set is additionally complemented with a data codebook, awarded a digital object identifier (doi: 10.5281/zenodo.3607986) and make available on the following link: https://doi.org/10.5281/zenodo.3607986.

Reviewers' comments:

R1. In this paper, the authors investigated the data for evidence-based decision-making by managers in hospitals and healthcare organization in Europe in 2019. From the discussion and conclusion, this paper emphasized the data how to be collected and used. This paper seems interesting for readers to understand Europe's managers why, what and how to use performance data. The authors may add the sections of suggestion and further study in this paper.

We proceeded accordingly and added a section with suggestions for further studies on the topic at the end of the discussion section of the paper (line numbers 524 – 541):

“To further advance the field, building on key findings of this work but also on its limitations, we propose a number of potential future research topics and methodological approaches. Looking at a more homogenous sample of healthcare managers’ profiles, and including other geographical settings - beyond Europe, while accounting for the influences of characteristics of specific healthcare systems, would lead to more generalisable results. We also suggest unpacking the reasons that hinder a more wide-spread use of performance data in decision-making on the operational level and doing so by looking at the topics of the lack of confidence in (underlying) data, timeliness, appropriateness and actionability of metrics as well as the skills needed to effectively use (including communicate) the data. Additionally, exploring the drivers of reported differences between privately and publicly-owned healthcare organisations, in their use of performance data, could optimise cross-learning opportunities. When looking into skills, crucial for middle managers in order to enhance the use of data, we propose paying special attention to benchmarking, as an organisational learning tool, which currently seems underused. Based on our experience from this study, we also suggest study designs and research modalities that do not only focus on the ability to ask questions. In line with (participatory) action research methodologies, future research should also try to engage participants into a discussion, validating survey-based data and eliciting further contextual information and even disagreements, thus deepening our understanding of what drives effective and efficient use of performance data for decision-making.”

Once again, thank you for giving us the opportunity to submit a revised version of the manuscript.

Yours sincerely,

Damir Ivankovic and co-authors

---

## [Editor Report · Decision Letter 1]

23 Mar 2020

Why, what and how do European healthcare managers use performance data? Results of a survey and workshop among members of the European Hospital and Healthcare Federation

PONE-D-20-01236R1

Dear Dr. Ivankovic,

We are pleased to inform you that your manuscript has been judged scientifically suitable for publication and will be formally accepted for publication once it complies with all outstanding technical requirements.

With kind regards,

Itamar Ashkenazi

Academic Editor

PLOS ONE

---

## [Editor Report · Acceptance letter]

26 Mar 2020

PONE-D-20-01236R1 

Why, what and how do European healthcare managers use performance data? Results of a survey and workshop among members of the European Hospital and Healthcare Federation 

Dear Dr. Ivankovic:

I am pleased to inform you that your manuscript has been deemed suitable for publication in PLOS ONE. Congratulations! Your manuscript is now with our production department. 

With kind regards,

on behalf of

Dr. Itamar Ashkenazi 

Academic Editor

PLOS ONE